# Risk-Averse Active Sensing for Timely Outcome Prediction under Cost Pressure

**Yuchao Qin**
Univerisity of Cambridge, UK

**Mihaela van der Schaar**
University of Cambridge, UK
Alan Turing Institute, UK

**Changhee Lee**
Chung-Ang University, South Korea

## Abstract

Timely outcome prediction is essential in healthcare to enable early detection and intervention of adverse events. However, in longitudinal follow-ups to patients' health status, cost-efficient acquisition of patient covariates is usually necessary due to the significant expense involved in screening and lab tests. To balance the timely and accurate outcome predictions with acquisition costs, an effective active sensing strategy is crucial. In this paper, we propose a novel risk-averse active sensing approach RAS that addresses the composite decision problem of *when* to conduct the acquisition and *which measurements* to make. Our approach decomposes the policy into two sub-policies: acquisition scheduler and feature selector, respectively. Moreover, we introduce a novel risk-aversion training strategy to focus on the underrepresented subgroup of high-risk patients for whom timely and accurate prediction of disease progression is of greater value. Our method outperforms baseline active sensing approaches in experiments with both synthetic and real-world datasets, and we illustrate the significance of our policy decomposition and the necessity of a risk-averse sensing policy through case studies.

## 1 Introduction

Timely decision-making through accumulated observation history has attracted significant attention in the machine learning community, with broad impact and applications in healthcare [8]. Consider a typical decision-making scenario in which an agent employs a *sensing policy* to actively collect diagnosis-related information from an underlying feature trajectory. In the presence of *cost pressure*, the goal of the agent is to achieve timely and accurate predictions of a time-varying outcome of interest based on the sensing history, i.e., feature observations accumulated over time, while maintaining a reasonable expense of feature acquisition. At each decision step, we refer to the problem of deciding *when to make new acquisitions* and *which features to measure* as *active sensing* [2, 24]. The optimal sensing policy can be achieved by negotiating the subjective trade-off between outcome prediction (accuracy and timeliness) and acquisition cost over time.

A special class of the active sensing problem has been extensively explored in the optimal stopping literature [2, 6, 15]. In optimal stopping, the agent focuses on the timely diagnosis of a terminal status, where observation ceases immediately after a confident diagnosis can be made [1, 5]. Thereby, the acquisition cost in optimal stopping is typically considered as the time span of the information collection process. Recently, Jarrett and van der Schaar [9] introduced a novel sensing framework to allow for the consideration of measurement cost of individual feature variables in optimal stopping, enabling better scheduling of the acquisition sequences. However, their approach still focuses on

a static terminal status and is unsuitable for general purpose active sensing where the outcome of interest varies over time.

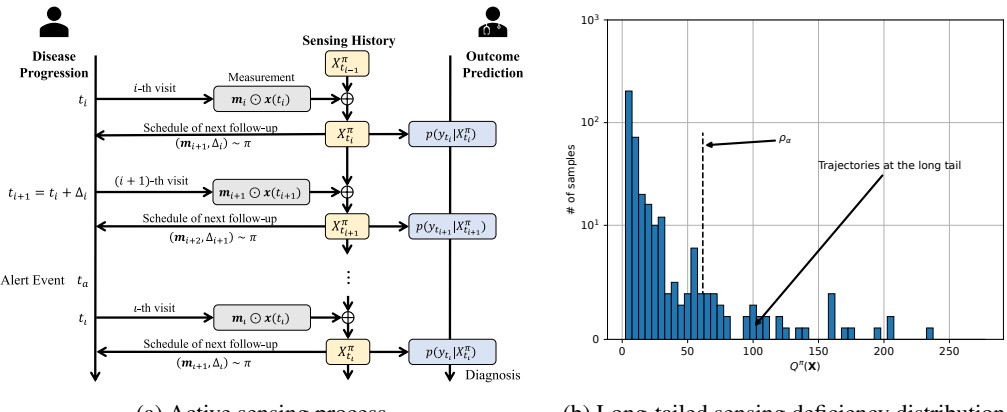

(a) Active sensing process        (b) Long-tailed sensing deficiency distribution

Figure 1: Overview of active sensing strategies. Active sensing in the dynamical setting (a) involves two distinct decisions on: what features to measure; and when to perform next acquisition. The trade-off between diagnosis accuracy and acquisition cost usually yields a (b) long-tailed sensing deficiency distribution. The sensing deficiency $Q^\pi(\boldsymbol{X})$ accumulates the weighted sum of the diagnosis error and measurement cost over time. Higher sensing deficiency values indicate failures in the sensing process.

In this paper, we focus on more general healthcare scenarios where timely diagnosis and intervention need to be achieved through active sensing under certain budget. The budget serves as the source of cost pressure and prioritize safe and low-cost measurements in the sensing process. For instance, the management of Alzheimer's disease (AD) requires continuous monitoring and staging of disease progression, which is crucial for early detection and intervention of high-risk patients [8]. However, certain measurements involved in diagnosing AD are notably expensive. A typical example is positron emission tomography (PET) scans. PET scans provide precise information about AD-associated proteins in the brain and are widely used for monitoring AD progression [3]. Nevertheless, the high accuracy of PET scans compared to cognitive tests is accompanied by greater financial costs as well as potentially harmful exposure to radiation [7]. A desirable active sensing policy should adaptively balance measurement costs and diagnostic accuracy at different disease stages to provide timely diagnosis and intervention under cost pressure. Such property is not only useful in healthcare but also other applications such as delay-sensitive wireless communication [14].

Another crucial but relatively unexplored aspect of active sensing is the consideration of *tail risk*. To illustrate this concept, let's examine the prediction of AD progression. In active sensing, the trade-off between diagnosis error and acquisition cost is usually achieved through the minimization of their weighted sum [9, 23, 24]. As depicted in Figure 1(b), the weighted objective ($Q^\pi(\boldsymbol{X})$) typically exhibits a long-tailed distribution with a notable concentration of *high-value* patients at the tail side. We consider these patients to be of higher value in the sensing task for two main reasons. Firstly, accurate diagnosis of many high-risk patients heavily relies on biomarkers obtained through expensive PET scans such as fluorodeoxyglucose (FDG) PET and florbetapir (AV45) PET [7, 10]. A vanilla sensing policy optimized for the entire population would easily fail to arrange enough PET scans for these high-risk patients due to the high costs, leading to degraded sensing performance (the long tail as shown in Figure 1(b)). In the meantime, the majority of patients tend to remain in stable conditions. Their disease stages can usually be diagnosed even with very sparse acquisitions over time. A biased sensing policy may overemphasize the importance of high-risk patients and order frequent PET scans for all patients to avoid false negatives in its diagnosis, which contributes to the long-tailed distribution from another perspective. To avoid such situations, an active sensing strategy must be *risk-averse* such that it can effectively cater to the needs of both the majority of stable patients and the high-risk subgroups requiring frequent follow-ups, ensuring timely and accurate detection of disease progression under cost pressure.

**Contributions.** In this paper, we propose a novel continuous-time active sensing approach to address the aforementioned desiderata. Our sensing approach adaptively balances measurement costs and diagnostic accuracy by factorizing sensing policy into two sub-policies: *acquisition scheduler* which

determines *when to conduct feature acquisitions* (i.e., the time interval between adjacent follow-ups), and *feature selector* that decides *which observations to make*. Furthermore, our sensing approach adopts the *risk-averse* strategy and focuses on the trade-offs happen at the long tail, where accurate diagnosis becomes particularly crucial. By dynamically identifying patient trajectories located at the tail side, we optimize our sensing strategy using the conditional value-at-risk (CVaR) loss to provide guarantees on timely and accurate prediction for high-risk patients while avoiding biased decisions for patients with stable conditions.

We validate our approach through experiments on synthetic and real-world datasets, where the sensing performance on high-risk patients is significantly impacted by their relatively small cohort size. With the consideration of acquisition costs, our method is able to adaptively identify the patient subgroups at the long tail of sensing deficiency distribution and achieves improved diagnostic performance compared to the state-of-the-art active sensing benchmarks.

## 2 Active Sensing in Practice

In this section, with a focus on healthcare applications, we introduce a new active sensing approach, which we refer to as *Risk-averse Active Sensing* (RAS), to solve the decision-making problem of when to conduct feature acquisitions and which features to observe considering prediction accuracy, timeliness, and cost pressure. We will start by providing a general framework for active sensing problems.

### 2.1 Preliminaries: Active Sensing

**Notation.** Let $\boldsymbol{x}(t) \in \mathbb{R}^d$ be the characteristic of a patient at time $t \in [0, T]$ which is a collection of $d$ (costly) observable time-varying covariates about an underlying disease progression of our interest.[1] $\boldsymbol{X}_t = \{\boldsymbol{x}(\tau) | \tau \in [0, t]\}$ represents a temporal trajectory of observable features until time $t \le T$. For ease of description, we will occasionally abuse the notation and use $\boldsymbol{X}$ to denote the entire patient trajectory $\boldsymbol{X}_T$. Let $y_t \in \mathcal{Y}$ be the outcome of our interest which represents the underlying health status of a patient (e.g., diagnosis, adverse outcome, etc.) at time $t$. We assume the joint distribution $p(\boldsymbol{X}_t, y_t)$ from which the trajectory $\boldsymbol{X}_t$ and the outcome $y_t$ are drawn. Hereafter, we focus our description on $K$-class classification tasks, i.e., $\mathcal{Y} = \{1, 2, \ldots, K\}$.

We denote an active sensing policy as $\pi$. In the sequential outcome prediction task, for each patient $\boldsymbol{X}$, policy $\pi$ arranges a set of follow-up times $\{t_1, t_2, \ldots, t_I\} \subset [0, T]$ and determines the subsets of features $\boldsymbol{m}_i$ to be measured at each time step $t_i$, where $\boldsymbol{m}_i$ is a binary vector. The collected observations of patient trajectory $\boldsymbol{X}$ until time $t$ form a sensing history $X_t^{\pi}$. It is worth noting that $X_t^{\pi}$ is piece-wise constant over time as it is only updated at discrete follow-ups. At each step, the sensing policy $\pi$ shall generate a diagnosis of the patient by approximating the conditional $p(y_t | X_t^{\pi})$.

**Dataset.** For the active sensing task, we assume access to an *observational dataset* $\mathcal{D}$ containing electrical health records (EHRs) of $N$ patients, i.e., $\mathcal{D} = \{X_n = (t_i, \boldsymbol{x}(t_i), y_{t_i})_{i=1}^L\}_{n=1}^N$. The record of each patient consists observations collected from $L$ follow-ups. It is worth noting that the record $X_n \in \mathcal{D}$ is equivalent to a sensing history.

Following the convention in active sensing literature [9, 24], we consider the *cost of measurement* on each time-varying observable feature to be time-invariant and thus can be represented as a *cost vector* $\boldsymbol{c}$, where the $j$-th element, denoted as $(\boldsymbol{c})_j$, is the cost for measuring the $j$-th time-varying feature in vector $\boldsymbol{x} \in \boldsymbol{X}$. The central goal of active sensing is then to balance between the outcome estimation accuracy based on sensing history $X_t^{\pi}$ and the accumulated acquisition costs over time.

**Challenges.** In practice, there are two major challenges that confound the active sensing problem:

1. **Adaptive follow-up intervals.** Most existing active sensing policies assume a constant follow-up interval. However, in real-world healthcare settings, the desirable acquisition intervals usually change along with disease progression. Sensing policy with constant decision intervals creates redundant follow-ups and may induce extra costs for feature acquisition.

---

[1]In general, observation of a patient's underlying disease progression ends at some time $T > 0$ due to death or lost to follow-up.

2. **Failures at the long tail.** Vanilla active sensing policies are optimized for the entire population. For diseases like AD, the imbalanced proportion of high-risk and stable patients may lead to over-conservative sensing policies that make no observations. Similarly, sensing policies targeted on high-risk patients may generate biased decision with higher acquisition costs for patients in stable conditions. Both situations lead to failures for samples at the long tail of sensing deficiency distribution.

In the following, we first introduce a generalized sensing policy $\pi$ with the decomposition of decisions on follow-up interval and feature selection. Then, we formulate the active sensing task into a risk-averse optimization problem, the solution to which addresses the second challenge.

## 2.2 Decomposing the Active Sensing Policy

Vast research in active sensing has assumed a constant follow-up interval between two consecutive patient visits and has primarily focused on the sequential feature selection problem, determining which feature variables to measure [6, 24]. However, considering the dynamic and complex nature of disease progression, sensing policies with dynamic follow-up intervals are broadly desired in real-world applications. In this paper, we propose a generalized formulation of the sensing policy, i.e., $\pi = (\pi_m, \pi_\Delta)$, where the *acquisition scheduler* $\pi_\Delta$ determines the time for next follow-up, the *feature selector* $\pi_m$ determines which features to measure during the subsequent visit.

Consider the $i$-th follow-up of patient $\boldsymbol{X}$ at time $t_i$. The next follow-up time $t_{i+1}$ is firstly determined by acquisition scheduler $\pi_\Delta$ as $t_{i+1} = t_i + \Delta_i, \Delta_i \sim \pi_\Delta(X_{t_i}^\pi)$. Then, at time $t_{i+1}$, the feature selector $\pi_m$ determines which subset of patient covariates to measure using a binary vector $\boldsymbol{m}_{i+1} \sim \pi_m(X_{t_i}^\pi), \boldsymbol{m}_{i+1} \in \{0,1\}^d$. The new observation is calculated as $\boldsymbol{m}_{i+1} \odot \boldsymbol{x}(t_{i+1})$, where the $j$-th element

$$(\boldsymbol{m}_{i+1} \odot \boldsymbol{x}(t_{i+1}))_j = \begin{cases} (\boldsymbol{x}(t_{i+1}))_j & \text{if } (\boldsymbol{m}_{i+1})_j = 1, \\ * & \text{otherwise.} \end{cases}$$

## 2.3 Towards Risk-Averse Sensing Policy

**Trade-off between accuracy and costs.** Consider the $i$-th follow-up at time $t_i$. The feature selection is determined by $\boldsymbol{m}_i \sim \pi_m(\tilde{X}_{t_{i-1}}^\pi)$. The acquisition cost is evaluated as $r_m(\boldsymbol{m}_i) = \boldsymbol{c}^\top \boldsymbol{m}_i$, where $\boldsymbol{c}$ is the cost vector. The updated sensing history $X_{t_i}^\pi = X_{t_{i-1}}^\pi \cup \{\boldsymbol{m}_i \odot \boldsymbol{x}(t_i)\}$ is then used to determine the acquisition interval $t_{i+1} - t_i = \Delta_i \sim \pi_\Delta(X_{t_i}^\pi)$. Assume a diagnosis error function $r_y(X_{t_i}^\pi, \Delta_i)$ that evaluates the mismatch between $p(y_t|\boldsymbol{X}_t)$ and $p(y_t|X_t^\pi)$ during the interval $\Delta_i$. The trade-off between diagnosis accuracy and measurement costs of the $i$-th interval is defined as the following reward signal

$$r(X_{t_i}^\pi, \boldsymbol{m}_i, \Delta_i) = r_m(\boldsymbol{m}_i) + \lambda r_y(X_{t_i}^\pi, \Delta_i) \tag{1}$$

where $\lambda > 0$ is a coefficient chosen to balance the two terms.

**Sensing deficiency.** We define the sensing deficiency of policy $\pi$ along a patient trajectory $\boldsymbol{X}$ as the *expected cumulative reward* as shown in (2).

$$Q^\pi(\boldsymbol{X}) = \mathbb{E}_{t_1,t_2,\ldots,t_I,\boldsymbol{m}_1,\boldsymbol{m}_2,\ldots,\boldsymbol{m}_I \sim \pi}\left[\sum_{i=1}^I \gamma^{i-1} r(X_{t_i}^\pi, \boldsymbol{m}_i, \Delta_i)\right], \tag{2}$$

where discount factor $\gamma \in (0, 1]$ is used to tackle long trajectories. In this paper, we take $\gamma = 0.99$.

**Trajectories at the long tail.** As mentioned earlier, one major challenge for active sensing in practice is the potential sensing failures for patients at the tail of sensing deficiency distribution. Since the distribution tail of $Q^\pi(\boldsymbol{X})$ changes along with the updates of policy $\pi$, in this paper, we propose to optimize the sensing policy over a dynamic subset of patient trajectories located at the long tail. These tail trajectories can be identified through the upper $\alpha$ quantile $\rho_\alpha$, where $\mathbb{E}_{\boldsymbol{X}}[\mathbb{I}(Q^\pi(\boldsymbol{X}) \geq \rho_\alpha)] = \alpha$. Assume the marginal trajectory distribution $p(\boldsymbol{X})$, we denote with $S_\alpha^\pi = \{\boldsymbol{X}|\boldsymbol{X} \sim p(\boldsymbol{X}), Q^\pi(\boldsymbol{X}) \geq \rho_\alpha\}$ the set of tail trajectories under policy $\pi$.

Conditioned on the quantile factor $\alpha \in [0, 1]$, the sensing deficiency of trajectories in $S_\alpha^\pi$ is considered as the value at risk that needs to be improved. Based on the evaluation of CVaR in sensing deficiency distribution, we formulate our active sensing task as a risk-averse optimization problem.

$$\underset{\pi}{\text{minimize}} \ \ \text{CVaR}_\alpha \triangleq \mathbb{E}_{\boldsymbol{X} \in S_\alpha^\pi}[Q^\pi(\boldsymbol{X})]. \tag{3}$$

Solutions to problem (3) are considered risk-averse since they reduce the risk of sensing failures in the worst scenarios ($S_\alpha^\pi$). Our proposed risk-averse sensing problem has two major distinctions from the class of robust adversarial reinforcement learning (RL) problems (e.g., [18]): i) we directly optimize the value at risk $\text{CVaR}_\alpha$; and ii) the adversarial agent is replaced by the dynamic tail trajectory subset $S_\alpha^\pi$. As such, our proposed problem requires no online interactions with the data generation process and thus can avoid the sensing policy $\pi$ from being biased toward unrealistic or out-of-distribution trajectories generated by simulators.

## 3 Method: Risk-Averse Active Sensing

Solution to the risk-averse active sensing problem in (3) given observational data is faced with three unique challenges. First, we only have access to observational data that consist of *discrete* measurements of (continuous) patient trajectories. Second, we need access to a baseline conditional outcome estimator, $f_P(y_t|\boldsymbol{X}_t)$, to estimate the diagnosis error $r_y$ in (1). Third, the solution of (3) heavily relies on the appropriate construction of the subset, $S_\alpha^\pi$, which, by definition, depends on the sensing policy itself.

**Handling discrete and sparse observations.** Dataset $\mathcal{D}$ only contains discrete observations of patient trajectories. However, the adaptive acquisition intervals generated by sensing policy $\pi$ may require measurements of the underlying patient trajectories at time points not included in $\mathcal{D}$. To solve this problem, we adopt a linear interpolator $\mathcal{I}$ as a proxy to obtain estimations of these values. In the meantime, policy $\pi$ by nature generates sparse sensing history with partial observations of patient trajectories. To properly encode the missingness and adaptive observation intervals in sensing history $X_t^\pi$, we include the neural CDE as the base encoders in our approach. As enlightened in [11], CDE is a continuous-time generalization to recurrent neural network (RNN) and offers a natural and elegant approach to encode the information in sparse and irregular measurements of a patient trajectory.

Specifically, given a discrete sensing history $X_t^\pi$, we first convert it into a continuous trajectory $\boldsymbol{u}(\tau) = \mathcal{I}(\tau, X_t^\pi)$ via the interpolator $\mathcal{I}$. Then, CDE encodes the temporal changes in $X_t^\pi$ as the dynamics of a latent variable $\boldsymbol{z}(t) \in \mathbb{R}^l$ via

$$\boldsymbol{z}(t) = \boldsymbol{z}(t_0) + \int_{t_0}^{t} f_\phi(\boldsymbol{z}(\tau)) \mathrm{d}\boldsymbol{u}(\tau), \text{ for } \tau \in [t_0, t], \tag{4}$$

where $t_0 \geq 0$, $f_\phi : \mathbb{R}^l \mapsto \mathbb{R}^{l \times d}$ is a map with learnable parameters $\phi$, initial state $\boldsymbol{z}_0 = \boldsymbol{z}(t = 0)$ could be manually specified or learned from data.

**Timely and accurate predictions.** In this paper, we build CDE-based outcome estimator $f_P(t, X)$ to approximate the density $p(y_t|\boldsymbol{X}_t)$, where $X$ is a sensing history of trajectory $\boldsymbol{X}$. Details about $f_P$ can be found in the Appendix. Given a patient with record $X \in \mathcal{D}$, to provide timely and accurate predictions about the underlying disease progression, our sensing policy aims to minimize the mismatch between baseline estimate $f_P(\tau, X)$ and sensing history-based prediction $f_P(\tau, X_\tau^\pi)$. For this purpose, in time horizon $\tau \in [t_i, t_{i+1})$, the diagnosis error function for the $i$-th follow-ups is computed as the *cumulative mismatch*:

$$r_y(X_{t_i}^\pi, \Delta_i) \triangleq \int_{t_i}^{t_{i+1}^-} D_{\text{JS}}(f_P(\tau, X_{t_i}^\pi) \| f_P(\tau, X)) \mathrm{d}\tau, \tag{5}$$

where $D_{\text{JS}}$ is the JS divergence, $t_{i+1} = t_i + \Delta_i$.

The distinction from previous works that utilize a single-step mismatch – such as at a pre-specified time interval [24] – is that (5) provides a precise proxy to evaluate how timely and accurate the predictions are by gauging the impact with respect to the acquisition interval $\Delta_i$, which is a desired property. Adverse clinical outcomes of the underlying disease progression are likely to occur when the outcome probability, $p(y_t|\boldsymbol{X}_t)$, hits a certain threshold, often accompanied by a sudden rise or fall. Making too early or too late detection of such a rise/fall based on the sensing history, i.e., $p(y_t|X_t^\pi)$, will drastically increase the cumulative mismatch in (5).

**Penalty on invalid visits.** We define the follow-up at time $t_{i+1}$ as an invalid visit if no patient covariates are measured, i.e., $\boldsymbol{m}_i = \boldsymbol{0}$. Allowing invalid visits potentially yields degenerated active sensing policy and makes the policy decomposition meaningless. To prevent such situations, we

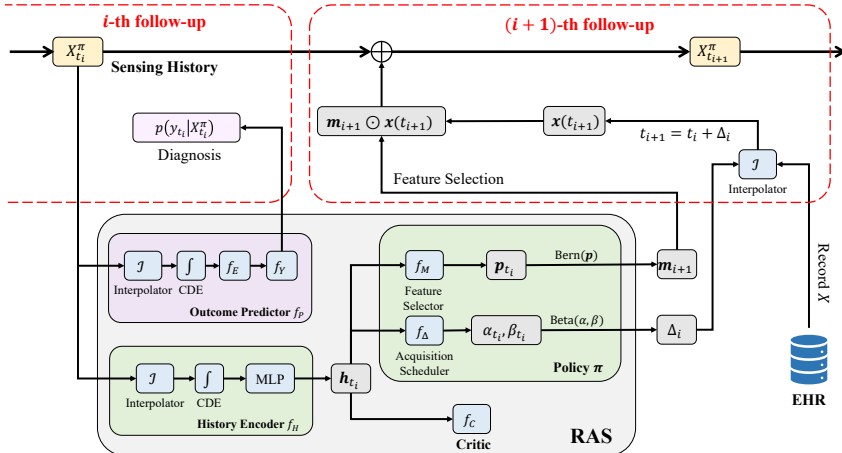

Figure 2: The model structure of RAS.

extend the acquisition cost function $r_m$ as $r_m(\boldsymbol{m}) = \boldsymbol{c}^\top \boldsymbol{m} \cdot \mathbb{I}(\boldsymbol{m} \neq \boldsymbol{0}) + \nu \cdot \mathbb{I}(\boldsymbol{m} = \boldsymbol{0})$, where $\nu \geq \boldsymbol{c}^\top \boldsymbol{1}$ penalizes the invalid visits scheduled by the sensing policy $\pi$.

**Network structure.** Similar to the predictor $f_P$, the neural CDE and interpolator $\mathcal{I}$ are also employed in our proposed policy network $\pi$ to construct an embedding of the discrete sensing history $X_t^\pi$. As illustrated in Figure 2, our sensing policy $\pi$ (parameterized by $\theta$) is constructed with the following three networks:

- The *history encoder*, $f_H$, is a neural CDE encoder that maps sensing history $X_t^\pi$ into a summary vector $\boldsymbol{h}(\tau) = f_H(\tau, \mathcal{I}(X_t^\pi))$.

- The *feature selector*, $f_M$, is a sub-policy network that governs the decision about which subset of patient features to be measured. In a follow-up, $f_M$ takes a summary vector $\boldsymbol{h}(\tau)$ as input and outputs a vector $\boldsymbol{p}(\tau) = f_M(\boldsymbol{h}(\tau))$ that governs the element-wise Bernoulli distribution to generate a binary feature selection vector $\boldsymbol{m}$, i.e., $\boldsymbol{m}_{\tau'} \sim \text{Bern}(\boldsymbol{p}(\tau))$ for next follow-up at $\tau'$.

- The *acquisition scheduler*, $f_\Delta$, is a sub-policy network that determines the time for next follow-up. Particularly, after each follow-up, it takes the summary vector $\boldsymbol{h}(\tau)$ as input and generates two scalar parameters $\alpha, \beta = f_\Delta(\boldsymbol{h}(\tau))$ of a Beta distribution, i.e., $\xi \sim \text{Beta}(\alpha, \beta)$. The realizations of the Beta distribution are then used to generate the time interval $\Delta \in [\Delta^{\min}, \Delta^{\max}]$ via $\Delta = \Delta^{\min} + (\Delta^{\max} - \Delta^{\min}) \cdot \xi$, where $\Delta^{\min}$ and $\Delta^{\max}$ are the minimum and maximum allowed intervals between two follow-ups, respectively. We set $\alpha, \beta \geq 1$ to ensure that the time interval follows a unimodal distribution.

**Policy gradient.** Since the diagnostic error function in (5) is non-differentiable, to *back-propagate* gradients from the objective $\text{CVaR}_\alpha = \mathbb{E}_{\boldsymbol{X} \in S_\alpha^\pi}[Q^\pi(\boldsymbol{X})]$ through the active sensing process, we apply the advantage-based approach [21, 22] to estimate the policy gradient of $\pi$ as the following:

$$\nabla_\theta \text{CVaR}_\alpha = \mathbb{E}_\pi \left[ \frac{1}{|S_\alpha^\pi|} \sum_{s=1}^{|S_\alpha^\pi|} \sum_{i=1}^{I} \left( R(X_{s,t_i}^\pi) - R_C(X_{s,t_i}^\pi) \right) \cdot \nabla_\theta \log \pi(\boldsymbol{m}_{s,i}, \Delta_{s,i}) \right], \quad (6)$$

where $\theta$ is the collection of learnable parameters of policy $\pi$. Given the $s$-th patient in $S_\alpha^\pi$, the cumulative reward $R(X_{s,t_i}^\pi) = \sum_{\iota=i}^{I} \gamma^{\iota-i} r(X_{s,t_\iota}^\pi, \boldsymbol{m}_\iota, \Delta_\iota)$ is an empirical evaluation of the sensing deficiency in (2) for the corresponding sub-trajectory in horizon $[t_i, T]$. Following the actor-critic framework, we utilize a baseline sensing deficiency estimation $R_C(X_{t_i}^\pi)$ in (6) to reduce the variance of policy gradient estimation [21].

The baseline estimation $R_C(X_t^\pi)$ is obtained from a *critic* network $f_C$ (parameterized by $\psi$). Taking the embedding $\boldsymbol{h}(t) = f_H(t, \mathcal{I}(X_t^\pi))$ from the history encoder $f_H$ as input. The critic $f_C$ is trained

to minimize the following mean squared error (MSE) loss.

$$\mathcal{L}_C = \mathbb{E}_\pi \left[ \frac{1}{|S_\alpha^\pi|} \sum_{s=1}^{|S_\alpha^\pi|} \sum_{i=0}^{I} (R(X_{n,t_i}^\pi) - R_C(X_{n,t_i}^\pi))^2 \right]. \tag{7}$$

---

**Algorithm 1** Risk-averse active sensing

---

**Input:** dataset $\mathcal{D}$, predictor $f_P$, quantile $\alpha$, learning rates $\mathrm{lr}_\pi$, $\mathrm{lr}_c$.
**Initialize:** learnable parameters $\theta$ (of $\pi$), $\psi$ (of $f_C$).
**for** $k = 1$ **to** $K$ **do**
   **if** $k - 1 \mod M = 0$ **then**
      $\rho_\alpha \leftarrow \texttt{quantile}(Q^\pi, \mathcal{D}, \alpha)$
      $S_\alpha^\pi \leftarrow \{X_n | Q^\pi(X_n) \geq \rho_\alpha, X_n \in \mathcal{D}\}$
   **end if**
   $\theta \leftarrow \theta - \mathrm{lr}_\pi \nabla_\theta \mathrm{CVaR}_\alpha(S_\alpha^\pi)$
   $\psi \leftarrow \psi - \mathrm{lr}_c \nabla_\psi \mathcal{L}_C(S_\alpha^\pi)$
**end for**
**Output:** risk-averse policy $\pi$ with parameters $\theta$

---

Given the baseline predictor $f_P$, the sensing deficiency $Q^\pi(\boldsymbol{X})$ of policy $\pi$ is optimized over the subset $S_\alpha^\pi$ of tail trajectories as illustrated in Algorithm 1. Specifically, parameters $\theta$ of policy $\pi$ are updated based on the policy gradient $\nabla_\theta \mathrm{CVaR}_\alpha$ in (6), and the critic network $f_C$ is trained with loss function $\mathcal{L}_C$ to provide appropriate baselines for the sensing policy $\pi$. The collection of patient trajectories in $S_\alpha^\pi$ is iteratively updated every $M$ training epochs such that it can stay informative about the sensing deficiency distribution. Note that the learning rates of critic $f_C$ and policy $\pi$ are intentionally set to be different to facilitate convergence.

## 4   Related Work

We first provide in Table 1 an overview of the distinction between our proposed approach and other related works in active sensing literature. A special class of active sensing task is formulated under the optimal stopping framework. With a set of explicitly defined terminal states, optimal stopping approaches focuses on achieving confident diagnosis of these states while minimizing the time span of the sensing process [1, 6, 15]. The consideration of information acquisition cost was introduced recently into optimal stopping in [9]. Similarly, the trade-off between diagnosis accuracy and costly measurement has been studied in the more general sequential prediction tasks [23, 24]. In this paper, we propose a novel sensing policy decomposition and extend the active sensing analysis into continuous-time settings with adaptive follow-up intervals and highly sparse sensing histories.

Table 1: Comparison of active sensing approaches. Typical active sensing approaches are compared based on the problem formulation, sensing history representation, the modelling of acquisition intervals as well as the following key concerns highlighted in this paper: (1) supports continuous-time patient trajectory. (2) considers cost pressure on features acquisition. (3) achieves decomposition of feature selection and acquisition scheduling. (4) provides worst-case performance guarantees with a risk-averse policy.

| Method | Problem Class | History Embedding | Acquisition Interval | (1) | (2) | (3) | (4) |
|---|---|---|---|---|---|---|---|
| Ahmad and Yu [1] | Optimal Stopping | Bayesian Update | Fixed ($\Delta = \tilde{\Delta}$) | ✗ | ✗ | ✗ | ✗ |
| Jarrett and van der Schaar [9] | Optimal Stopping | Bayesian Update | Fixed ($\Delta = \tilde{\Delta}$) | ✗ | ✓ | ✗ | ✗ |
| Yoon et al. [23, 24] | Sequential Prediction | RNN | Fixed ($\Delta = \tilde{\Delta}$) | ✗ | ✓ | ✗ | ✗ |
| RAS (Ours) | Sequential Prediction | Neural CDE | Adaptive ($\Delta \sim \pi_\Delta$) | ✓ | ✓ | ✓ | ✓ |

**Temporal feature selection.** As pointed out by Yoon et al. [24], active sensing can be considered as the temporal generalization of feature selection tasks [13, 19]. In each decision step, the sensing policy is expected to select a set of important feature variables to make the prediction. The resulting sensing history marks critical time points and key feature variables for accurate outcome estimation. However, existing active sensing literature primarily focuses on the feature selection perspective and usually assumes a fixed follow-up interval for every patient trajectory [1, 24]. A fixed acquisition

interval can lead to delayed diagnosis or extra costs due to redundant feature measurements. As mentioned earlier, a desirable sensing policy shall achieve flexible follow-up intervals for each patient such that the shifts in underlying disease stages can be properly addressed.

**Risk aversion.** Conventional active sensing strategies typically consider the diagnosis accuracy to be equally important for all trajectories in a dataset. However, in many healthcare applications, there are needs to prioritize smaller subgroup of high-risk patients for whom timely detection of adverse outcomes is particularly valuable. As a central component of our proposed method, the concept of CVaR [20] enables the selective optimization for patient trajectories in the long tail of sensing deficiency distribution. Optimizing sensing policies over the corresponding tail set $S_\alpha^\pi$ yields a risk-averse active sensing strategy $\pi^*$ that achieves guaranteed diagnosis accuracy for high-risk patients. While risk-aware decision-making has been extensively explored in the field of RL [4, 25], to the best of our knowledge, our proposed method is the first risk-averse active sensing algorithm for adaptive feature acquisition in continuous-time settings.

## 5 Experiment

In the experiments, we evaluate the effectiveness of our proposed risk-averse active sensing approach RAS on both synthetic and real-world healthcare datasets.

**Synthetic dataset.** We construct a synthetic dataset $\mathcal{D}_S$ of $N = 2000$ samples. Every sample $X$ contains $L = 20$ discrete observations in $t \in [0, 2]$. Each observation yields a feature vector $\boldsymbol{x} = [x_1, x_2, x_3, x_4]^\top$. We set $x_1(t) = \min(1, (e^{w(t-\tau)} - w(t-\tau) - 1))\mathbb{I}(t \geq \tau)$, where $w$ follows a Gaussian mixture distribution of $0.8 \cdot \mathcal{N}(0.3, 0.1^2) + 0.2 \cdot \mathcal{N}(0.8, 0.1^2)$, $\tau \sim \text{Exp}(1.0)$. As a proxy of $x_1$, we set $x_2(t) = w(t-\tau)\mathbb{I}(t \geq \tau)$. Variables $x_3(t) = \sin(3t + \varrho)$ and $x_4(t) = \cos(3t + \varrho)$, where $\varrho \sim \mathcal{N}(0, 1^2)$. The outcome $y \in \{0, 1\}$ follows a Bernoulli distribution $y_t \sim \text{Bern}(p)$, and the likelihood for $y_t = 1$ is calculated as $p = e^{-3|1-x_1|^2}$. The cost vector is set to be $\boldsymbol{c} = [1.0, 0.1, 1.0, 1.0]^\top$, which allows $x_2$ to be measured as a cheap proxy of $x_1$ for outcome prediction.

**ADNI dataset.** The Alzheimer's Disease Neuroimaging Initiative[2] (ADNI) dataset includes records on AD progression of $N = 1002$ patients with regular follow-ups every six months. We consider the active sensing task to predict patient outcomes in the initial $L = 12$ follow-ups with four biomarkers from PET (FDG and AV45) and MRI (Hippocampus and Entorhinal) imaging, respectively. Based on the disease staging guideline [16, 17], target outcome $y_t \in \{\text{Normal, mild cognitive impairment (MCI), AD}\}$ at each time point is determined via the corresponding Clinical Dementia Rating scale Sum of Boxes (CDR-SB) score with cutoff thresholds at 0.5 and 4.5. To reflect the higher cost and harmful radiation exposure in PET scan, measurement costs for PET and MRI biomarkers are set to be 1.0 and 0.5, respectively.

**Experiment setup.** We first fit the outcome estimator $f_P$ on each dataset with 64/16/20 train/validation/test splits and find the optimal estimator with the best prediction accuracy given sparse observation of test samples as inputs. Then, we freeze the parameters of the optimal $f_P$ and evaluate the sensing performance of RAS and other baselines on the corresponding data split. We consider four baselines in the experiments: **FO**) predictor $f_P$ with dense sensing history; **ASAC**) [24]; **NLL**) adaptive sensing with negative log-likelihood (NLL) as diagnosis error function [12]; **AS**) RAS($\alpha = 1.0$) with constant acquisition interval. Further details of the experiment setup and parameter selection results can be found in the Appendix.

Table 2: Benchmark of sensing performance on the synthetic dataset $\mathcal{D}_S$.

| METHOD | ROC | PRC | COST | $d_{\delta=0.3}$ | $d_{\delta=0.5}$ | $d_{\delta=0.7}$ |
|---|---|---|---|---|---|---|
| FO | **0.680±0.000** | **0.655±0.000**$^\ddagger$ | 31.000±0.000 | 0.502±0.000 | 0.349±0.000 | 0.285±0.000 |
| ASAC | 0.605±0.096 | 0.559±0.080 | **0.460±1.078**$^\ddagger$ | 1.099±0.664 | 1.066±0.699 | 1.052±0.641 |
| AS | 0.671±0.001 | 0.614±0.001 | 4.501±0.497 | 0.577±0.029 | 0.522±0.012 | 0.479±0.015 |
| NLL | 0.636±0.023 | 0.588±0.016 | 2.968±0.774 | 0.993±0.131 | 0.974±0.141 | 0.975±0.147 |
| RAS (OURS) | 0.680±0.003 | 0.647±0.006 | 6.077±0.953 | **0.325±0.084**$^\ddagger$ | **0.264±0.086**$^\ddagger$ | **0.246±0.071**$^\ddagger$ |

The 95% confidence interval (CI) is evaluated with five different random seeds. Best result in each column are highlighted in **bold**. The marker $^\ddagger$ indicates p-value $p < 0.05$.

---

[2] https://adni.loni.usc.edu

**Timeliness and accuracy.** Timely and accurate outcome prediction given sparse observations of patient trajectories is essential for practical applications of active sensing. In the experiments, we report the area under the curve of receiving-operator characteristic (ROC) and area under the curve of precision-recall (PRC) as assessments of the prognostic value of active sensing policies. In the meantime, the timeliness of an active sensing policy $\pi$ is evaluated with $d_\delta$ – the mean absolute mismatch in diagnosis time of event $p(y_t = 1 | X_t^\pi) \geq \delta$ compared to ground truth on the test set. The benchmark results of RAS against four baselines on the two datasets are provided in Table 2 and Table 3, respectively.

With a fixed acquisition interval of $\Delta = 0.2$, FO achieves high predictive accuracy at the expense of highest acquisition cost. Despite having the smallest acquisition cost, ASAC suffers significantly from the low diagnosis accuracy and is unable to achieve timely diagnosis of alert events. In contrast, our method (RAS) has reasonable acquisition costs and best timeliness as illustrated in Table 2. The increase in average cost of RAS compared to the AS baseline could be linked to our focus on the tail set $S_\alpha^\pi$. The increased acquisition costs are necessary to achieve a significantly lower delay in alert event diagnosis for high-risk patients in $S_\alpha^\pi$. The improved timeliness of RAS over FO suggests that the predictor $f_P$ might achieve better generalization with sparse sensing history as inputs. Similarly, according to Table 3, our method RAS outperforms most other baselines in timeliness and cost efficiency except for FO. This is within expectation since the outcome estimations from $f_P$ are used as the reference for timeliness evaluation, and these estimations are generally close to the predictions made by the FO baseline.

Table 3: Benchmark of active sensing performance on ADNI dataset.

| METHOD | ROC | PRC | COST | $d_{\delta=0.1}$ | $d_{\delta=0.3}$ | $d_{\delta=0.5}$ |
|---|---|---|---|---|---|---|
| FO | **0.747±0.000**[‡] | **0.577±0.000**[‡] | 26.865±0.000 | **0.141±0.000**[‡] | **0.510±0.000**[‡] | **0.591±0.000**[‡] |
| ASAC | 0.521±0.160 | 0.352±0.103 | **0.043±0.186**[‡] | 0.527±0.000 | 3.008±3.610 | 3.581±0.000 |
| AS | 0.704±0.023 | 0.519±0.034 | 3.566±0.854 | 1.326±0.096 | 2.314±0.348 | 2.357±0.375 |
| NLL | 0.697±0.018 | 0.512±0.020 | 3.986±0.493 | 1.040±0.149 | 2.176±0.060 | 2.739±0.135 |
| RAS (OURS) | 0.730±0.007 | 0.560±0.012 | 8.614±1.157 | 0.820±0.096 | 1.370±0.227 | 1.192±0.176 |

The 95% confidence interval (CI) is evaluated with three different random seeds. The timeliness $d_\delta$ is measured based on the likelihood of developing AD. Best result in each column are highlighted in **bold**. The marker [‡] indicates p-value $p < 0.05$.

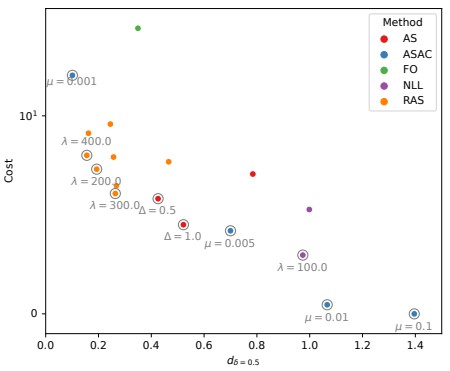

(a) Pareto front of sensing policies.

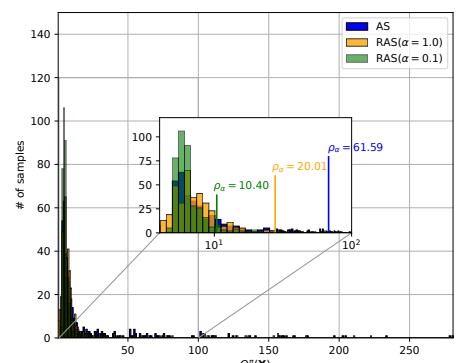

(b) Distributions of sensing deficiency $Q^\pi(\boldsymbol{X})$.

Figure 3: The impact of risk-aversion and adaptive acquisition scheduling on active sensing policies. Selection of trade-off coefficients, e.g., $\lambda$, can be performed by searching for the (a) Pareto front. Pareto optimal models belonging to the front are highlighted with circles. Their parameters are annotated in gray texts. Effectiveness of the risk-averse objective in (3) is reflected by the tail size of (b) $Q^\pi(\boldsymbol{X})$ distribution. The risk-neutral ($\alpha = 1.0$) version of RAS has degraded sensing performance for tail trajectories. The significantly worse performance of the AS baseline further highlights the benefit of having adaptive acquisition intervals.

**Trade-off between timeliness and acquisition costs.** The selection of model parameters could be difficult when two or more criteria are involved in the evaluation. Here, we illustrate the sensing performance of different policies on the synthetic dataset in Figure 3(a) and highlight the ones in the Pareto front with gray circles. These policies are considered Pareto optimal since their timeliness

$(d_{\delta=0.5})$ and average acquisition cost cannot be simultaneously improved by swapping parameters with other policies. Benefitted from the risk-averse training strategy, most sensing policies obtained via RAS are centered around the knee point of the Pareto front, which helps to explain the outstanding cost efficiency of RAS as reported in Table 2.

**Reshaping the sensing deficiency distribution.** To illustrate the effectiveness of the risk-averse objective in (3), we compare the empirical distribution of sensing deficiency $Q^\pi(\boldsymbol{X})$ of RAS with the ablations of risk-neutral sensing ($\alpha = 1.0$) and AS baseline ($\alpha = 1.0$, constant acquisition interval $\Delta = 1.0$) on the synthetic dataset. All three models are trained with the same trade-off coefficient $\lambda = 300$. As illustrated in Figure 3(b), RAS is able to effectively optimize the sensing performance for trajectories in the tail set $S_\alpha^\pi$ and reduces the upper $\alpha$ quantile of $Q^\pi(\boldsymbol{X})$ to $\rho_{\alpha=0.1} = 10.40$. Factor $\alpha = 1.0$ completely disables the risk-aversion training strategy in Algorithm 1. Thereby, a clear increase of sensing deficiency (quantile $\rho_{\alpha=0.1}$ grows from 10.40 to 20.01) is observed with the risk-neutral ablation of RAS. Similarly, without adaptive scheduling of acquisition intervals and risk-averse optimization strategies, the AS baseline illustrates the failure of conventional active sensing paradigms at the long tail of $Q^\pi(\boldsymbol{X})$ distribution.

# 6 Conclusion

In this paper, we introduce a novel risk-averse active sensing approach RAS to address the challenging continuous-time active sensing problem. Through effective decomposition of decisions on feature selection and acquisition intervals, our approach offers valuable insights on both feature importance and timeliness of patient follow-ups. The novel risk-aversion training strategy in RAS enables the prioritization of high-value patients at the long tail of sensing deficiency distribution and provides guarantees on diagnosis accuracy for worst-case scenarios. The effectiveness of our method is evaluated through experiments on synthetic and real-world healthcare datasets, where RAS outperforms all baseline active sensing approaches and achieves accurate and timely outcome diagnosis while maintaining reasonable costs in feature acquisition.

## Broader Impacts

The development of an advanced active sensing strategy is essential for precision medicine. Appropriate sensing policies can help clinicians to make tailored follow-up schedules for their patients and effectively utilize the costly and potentially harmful measurements on some important patient characteristics under certain levels of cost pressure. We note that the reduction in acquisition cost is usually associated with degraded diagnosis accuracy and delayed detection of some adverse clinical events, especially for high-risk patients that could benefit from more frequent follow-ups on disease progression. In this paper, we attempt to address this challenge through the risk-averse policy learning strategy in our proposed active sensing approach RAS and achieve improved sensing performance (accuracy and timeliness) on the entire population. Nevertheless, practical applications of active sensing approaches would require careful audits and assessments by human experts to avoid potential negative impacts. The sensing decisions from RAS are only suggestions optimized based on observational data and should not be directly applied without the evaluation of clinicians.

## Acknowledgements

Yuchao Qin was supported by the Cystic Fibrosis Trust. Changhee Lee was supported by the IITP grant funded by the Korea government (MSIT) (No.2021-0-01341, AI Graduate School Program, CAU). This work was supported by Azure sponsorship credits granted by Microsoft's AI for Good Research Lab.

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

# Appendix

## A.1 Notation table

A summary of major notations used in this paper is provided below.

- $\boldsymbol{x}$: Feature variable of the patient.
- $y$: Patient's outcome.
- $\boldsymbol{X}$: The continuous-time patient trajectory.
- $X_t^\pi$: The sensing history at time $t$.
- $\pi$: Sensing policy.
- $\pi_m$: Feature selector.
- $\pi_\Delta$: Acquisition scheduler.
- $\boldsymbol{m}$: Feature selection mask vector.
- $Q^\pi(\boldsymbol{X})$: Sensing deficiency.
- CVaR: Conditional value-at-risk.
- $\alpha$: Tail risk quantile.
- $T$: End time of observation.
- $I$: Number of observations in sensing history.
- $\mathcal{I}$: Interpolator.

## A.2 Limitation

The risk-averse policy learning strategy proposed in this paper relies on the correct identification of tail-risk patients. Data quality of the EHR datasets has direct impact on our method. Low quality dataset would lead to poor diagnosis accuracy of the baseline predictor $f_P$ which compromises the validity and usefulness of RAS in practice. In the meantime, in RAS, patients located at the long tail of sensing deficiency distribution are selected based on empirical evaluations of $Q^\pi(\boldsymbol{X})$. The selection results could be biased due to potential over-fitting of sensing policies on the training set. To tackle this issue, an "honest" and unbiased sensing performance evaluation on the validation set is necessary. However, the inference of acquisition costs from validation samples is beyond the main focus of this paper. We leave this challenge for our future work on risk-averse active sensing tasks.

**Code Availability.** The source code of RAS can be found in the two GitHub repositories listed below:

- The van der Schaar lab repo: `https://github.com/vanderschaarlab/cvar_sensing`
- The author's personal repo: `https://github.com/yvchao/cvar_sensing`

## A.3 Estimation of Time-Varying Outcome

A baseline outcome predictor $f_P$ is required in (5) of the manuscript to evaluate the timeliness and accuracy of a sensing policy. Following the model-based approach in [24], we introduce a network $f_P$ – which we refer to as the baseline *predictor* – to estimate the unknown conditional distributions of patient outcomes. Specifically, we construct the predictor $f_P$ with the neural CDE [11] model to flexibly handle sensing history with irregular observation intervals compared to the RNN-based counterparts such as [23, 24].

The predictor consists of four components, i.e., $f_P = f_Y \circ f_E \circ \text{CDE} \circ \mathcal{I}$. $\mathcal{I}$ is a linear interpolator. CDE is a neural CDE that takes the continuous trajectories as inputs. $f_E$ is a multi-layer perceptron (MLP) that takes the CDE embedding as input. $f_Y$ is an MLP with $\text{softmax}$ output layer for categorical outcomes. Utilizing the linear interpolator $\mathcal{I}^3$, both the discrete EHR record $X \in \mathcal{D}$ and sparse sensing history $X_t^\pi$ can be converted into continuous trajectories.

---

[3]We use Python package `torchcde` for interpolation.

The solution to the latent CDE provides embeddings of discrete time-series observations and the sensing history as $z_{n,t} = f_E(\text{CDE} \circ \mathcal{I}(X_{n,t}))$ and $z_{n,t}^\pi = f_E(\text{CDE} \circ \mathcal{I}(X_{n,t}^\pi))$, respectively. The corresponding conditional distributions are then estimated via the MLP as $f_P(X_{n,t}) = f_Y(z_{n,t})$ and $f_P(X_{n,t}^\pi) = f_Y(z_{n,t}^\pi)$, respectively.

The predictor $f_P$ is trained to minimize the following loss:

$$\mathcal{L}_P = \frac{1}{NL} \sum_{n=1}^{N} \sum_{l=1}^{L} \ell(y_{n,t_l}, f_Y(z_{n,t_l})) + \mu \ell(y_{n,t_l}, f_Y(z_{n,t_l}^{\pi^0})), \tag{8}$$

where $z_{n,t_l} = f_E(\text{CDE} \circ \mathcal{I}(X_{n,t_l}))$, $z_{n,t_l}^{\pi^0} = f_E(\text{CDE} \circ \mathcal{I}(X_{n,t_l}^{\pi^0}))$, and $\ell$ is the $K$-class cross-entropy based on the observed outcome $y_{n,t_l}$ and outputs of the predictor $f_P$. Here, we introduce the second term as a regularization for each patient trajectory with random dropouts. More specifically, we introduce a random acquisition strategy $\pi^0$ which randomly collects (sparse) discrete observations from the interpolated trajectories, i.e., $\mathcal{I}(X_n)$, and constructs auxiliary sensing histories with random dropouts, i.e., $X_{n,t_l}^{\pi^0}$. The strategy $\pi^0$ randomly select feature variables to keep in the observation history based on the drop rate. This regularization improves the generalization of the predictor by learning to estimate conditional distributions given randomly drawn sensing histories. Here, $\mu \geq 0$ is a coefficient that controls the strength of regularization.

## A.4 Parameter Selection

**Outcome predictor.** We consider the drop rate $p$ of the auxiliary observation strategy $\pi^0$ as the hyperparameter of the outcome predictor. For both the synthetic dataset and ADNI dataset, we select the drop rate of auxiliary strategy $\pi^0$ form the set of $p \in \{0.0, 0.3, 0.5, 0.7\}$ The optimal hyperparameter is selected based on the average accuracy of the outcome predictor on five test sets of random observations generated by an auxiliary observation strategy $\pi^0$ with drop rate $p = 0.7$. The hyperparameter selection result is as follows:

- **Synthetic dataset**: drop rate $p = 0.7$
- **ADNI dataset**: drop rate $p = 0.7$

**Sensing policy.** Then, we perform hyperparameter selection for the active sensing approaches. Among the trained outcome predictors, we select the one with highest ROC score on its test set as the shared predictor and keep its parameters frozen. The training and test set corresponding to the selected optimal outcome predictor are used for the training and evaluation of the active sensing methods.

In our experiments, the hyperparameters of each active sensing method are selected based on the cost efficiency, i.e., ROC / Cost, obtained on the test set.

For the synthetic data considered in our manuscript, we set the minimum and maximum allowed acquisition intervals as $\Delta^{\min} = 0.2, \Delta^{\max} = 1.0$, respectively. ASAC is not affected since it needs to directly work on original EHR records). The hyperparameters of each method are reported as follows:

- **ASAC**: coefficient for acquisition cost $\mu = 0.01 \in \{0.1, 0.01, 0.005, 0.001\}$.
- **AS**: acquisition interval $\tilde{\Delta} = 1.0 \in \{0.2, 0.5, 1.0\}$, shared predictor $f_P$ with RAS.
- **NLL**: coefficient for diagnostic error $\lambda = 100 \in \{100, 300\}$, shared predictor $f_P$ with RAS.
- **RAS**: the coefficient for diagnostic error $\lambda = 300 \in \{200, 250, 280, 300, 310, 320, 350, 400\}$, discount factor $\gamma = 0.99$, tail-risk quantile $\alpha = 0.1$, penalty for invalid visits $\nu = 10$.

For the ADNI data considered in our manuscript, we set the minimum and maximum allowed acquisition intervals as $\Delta^{\min} = 0.5, \Delta^{\max} = 1.5$, respectively. ASAC is not affected since it needs to directly work on original EHR records).

- **ASAC**: coefficient for acquisition cost $\mu = 0.1 \in \{0.1, 0.01, 0.005, 0.001\}$.
- **AS**: acquisition interval $\tilde{\Delta} = 1.5 \in \{0.5, 1.5\}$, shared predictor $f_P$ with RAS.

- **NLL**: coefficient for diagnostic error $\lambda = 200 \in \{100, 200, 300, 400\}$, shared predictor $f_P$ with RAS.

- **RAS**: the coefficient for diagnostic error $\lambda = 400 \in \{200, 250, 300, 350, 400, 450\}$, discount factor $\gamma = 0.99$, tail-risk quantile $\alpha = 0.1$, penalty for invalid visits $\nu = 10$.

All methods are trained with $K = 200$ iterations in the experiments. For RAS, we set the tail subset update interval $M = 10$ for Algorithm 1 in the manuscript.

**Adjustments in experiment results.** We have fixed some minor issues with model convergence when using CDE models. Specifically, the negative feedback is included in the CDE integration as follows:

$$\boldsymbol{z}(t) = \boldsymbol{z}(t_0) + \int_{t_0}^{t} (f_\phi(\boldsymbol{z}(\tau)) - 10^{-3}\mathrm{diag}(\boldsymbol{z}))\mathrm{d}\boldsymbol{u}(\tau), \text{for } \tau \in [t_0, t], \tag{9}$$

where $\mathrm{diag}$ creates a diagonal matrix from the input vector $\boldsymbol{z}$. Such negative feedback loop avoids the unlimited growth of CDE integration over time and helps with model convergence. This modification causes some minor changes in the experiment results. However, the updated experiment results are consistent with the previous versions and our conclusion is not affected by these changes.

**Additional results.** We will release more evaluation on real-world datasets on our GitHub repo.

