# OpenReview forum: "Risk-Averse Active Sensing for Timely Outcome Prediction under Cost Pressure"
_NeurIPS.cc/2023/Conference — NeurIPS 2023 poster_

### Official Review · Reviewer_d1SC · 2023-06-23

**Soundness:** 2 fair
**Presentation:** 2 fair
**Contribution:** 2 fair
**Rating:** 5
**Confidence:** 5

**Summary:**

This paper studies the problem of balancing timely and accurate outcome predictions with acquisition costs. To this end, a risk averse active sensing approach (RAS) is proposed that determines when to perform feature acquisition as well as which features to acquire. The proposed approach decomposes the policy into an acquisition scheduler that decides when to perform feature acquisition, and a feature selector that decides which features to acquire. A risk-averse training strategy is introduced to focus on high-risk patients. The proposed approach is evaluated on a synthetic dataset and a real-world dataset from Alzheimer's Disease Neuroimaging Initiative.

**Strengths:**

+ A continuous-time risk-averse active sensing approach is proposed to balance timely and accurate outcome predictions with acquisition costs.
+ The proposed approach optimizes timely and accurate prediction for tail-risk patients.
+ Experimental results on a synthetic and a real-world dataset are provided that illustrate the performance of the proposed approach compared to 3 baselines.

**Weaknesses:**

- Even though the problem of active sensing over time (also known as active state tracking) is well-studied, the related work section is very thin on references.
- Certain parts of the problem and solution description are unclear or left for the appendix.
- Certain notation definitions are missing.
- Certain decision choices are not justified.
- The structure of the proposed solution, as illustrated in Fig. 2, is not justified.
- Experiments are provided only on two datasets, out of which 1 is synthetic.

**Questions:**

1. The problem of active sensing over time (also known as active state tracking) is well-studied, but the related work section does not include any such references. An example set of references are provided below:
 - Krishnamurthy, V., 2002. Algorithms for optimal scheduling and management of hidden Markov model sensors. IEEE Transactions on Signal Processing, 50(6), pp.1382-1397.
- Krishnamurthy, V. and Djonin, D.V., 2007. Structured threshold policies for dynamic sensor scheduling—A partially observed Markov decision process approach. IEEE Transactions on Signal Processing, 55(10), pp.4938-4957.
- Krishnamurthy, V., 2013. How to schedule measurements of a noisy Markov chain in decision making?. IEEE Transactions on Information Theory, 59(7), pp.4440-4461.
- Zois, D.S. and Mitra, U., 2017. Active State Tracking With Sensing Costs: Analysis of Two-States and Methods for $ n $-States. IEEE Transactions on Signal Processing, 65(11), pp.2828-2843.
- Molloy, T.L. and Nair, G.N., 2021, June. Active trajectory estimation for partially observed Markov decision processes via conditional entropy. In 2021 European Control Conference (ECC) (pp. 385-391). IEEE.
- Atia, G.K., Veeravalli, V.V. and Fuemmeler, J.A., 2011. Sensor scheduling for energy-efficient target tracking in sensor networks. IEEE Transactions on Signal Processing, 59(10), pp.4923-4937.
- Beyer, C., Büttner, M., Unnikrishnan, V., Schleicher, M., Ntoutsi, E. and Spiliopoulou, M., 2020. Active feature acquisition on data streams under feature drift. Annals of Telecommunications, 75, pp.597-611.
-  Kossen, J., Cangea, C., Vértes, E., Jaegle, A., Patraucean, V., Ktena, I., Tomasev, N. and Belgrave, D., 2022. Active Acquisition for Multimodal Temporal Data: A Challenging Decision-Making Task. arXiv preprint arXiv:2211.05039.

A careful literature review and follow-up discussion is needed to place the current paper within the vast prior work on this area and clarify its novelty.

2. The problem studied in this paper also relates to quickest change detection, so it will be informative to include a discussion how the proposed approach contrasts/relates to this line of work. Some example references are provided:
 - Heydari, J. and Tajer, A., 2017, March. Quickest change detection in structured data with incomplete information. In 2017 IEEE International Conference on Acoustics, Speech and Signal Processing (ICASSP) (pp. 6434-6438). IEEE.
- Chaudhuri, A., Fellouris, G. and Tajer, A., 2021, July. Sequential change detection of a correlation structure under a sampling constraint. In 2021 IEEE International Symposium on Information Theory (ISIT) (pp. 605-610). IEEE.

3. Certain parts of the problem and solution description are unclear or left for the appendix. For example, the discussion on the training of the predictor, which is necessary to fully understand how the proposed approach works, is left in the appendix. In addition, it should be mentioned early on in Section 2.1 that a subset of features are selected when acquisition happens.

4. Certain notation definitions are missing or are confusing. For instance, D_{JS}(.) in Eq. (1) is not defined (I assume it is the Jensen-Shannon divergence). Furthermore, what is the relationship between T and I?  In Eq. (6), m and \Delta have 3 indices, not one as before, why? In Algorithm 1, A & B should be inputs, right? How are they selected in practice? What is their effect on the performance of the algorithm?

5. The structure of the proposed solution, as illustrated in Fig. 2, is not justified. Why this structure is adopted? Why these neural networks?

6.  Certain decision choices are not justified. For example, why the beta distribution is adopted for the policy?

7. Experiments are provided only on two datasets, out of which 1 is synthetic. These are not enough to justify the performance of the proposed approach. To make matters worse, three baselines are provided, out of which only 1 is from prior work. The results on real-world data are not impressive, and in fact comparable with the baselines, apart from the feature acquisition reduction. No statistical significance is provided. There is no justification of the synthetic dataset setting.

8. The usage of the terminology "cost pressure on patient trajectory acquisition" is unclear. Also, somewhere in the paper, it is mentioned that there is a budget on acquisition costs but I could not locate where this budget gets into the problem formulation.

**Limitations:**

The authors have addressed the limitations of the proposed approach and the potential negative societal impact of their work.

---

> ### Author Rebuttal · Authors · 2023-08-10
>
> We thank the reviewer for the helpful comments and efforts towards improving our manuscript. We provide responses in regard to the reviewer's concerns as follows.
>
> ## 1. Clarity
> - Following the reviewer's advice, we have moved discussions related to our problem formulation and solutions to the main manuscript.
> - A notation table is available in our response to Reviewer L7Er. Discussion of parameters $A$ and $B$ in Algorithm 1 can be found in Section A.5 of the Appendix.
> - Beta distribution $\mathrm{Beta}(\alpha, \beta)$ is adopted in our model because i) $\mathrm{Beta}(\alpha, \beta)$ is an uni-modal distribution when $\alpha,\beta>1$; ii) sampling from a Beta distribution is efficient, which helps to reduce the computational complexity.
> - "Budget" and "cost pressure". Our considered budget is a constraint on the accumulated feature acquisition cost, which creates the "cost pressure" on active sensing policies. For a practical solution to our sensing problem, we relax the budget constraint using a Lagrange approximation (second term of reward signal $r$ in Eq. (2)).
>
> ## 2. Related work
> In the active sensing task, our target is to balance the label prediction accuracy and feature acquisition cost. For this purpose, sensing policies should be able to make decisions on both when to perform new measurements and which feature subset to observe. We thank the reviewer for the detailed list of relevant literature. However, we found that many of these studies are incompatible with our considered active sensing setup. We provide analyses by paper category as follows.
>
> - Hidden Markov models (HMM) [1]: There are two possible ways to align the HMM formulation into our active sensing settings.
>     - Consider label $y$ as the state of a hidden Markov chain. The dependency between sensing history and label $y$ cannot be properly captured by the HMM structure, which makes these HMM-based methods unsuitable for our active sensing task. While the approach proposed in [2] can solve the problem of when to observe in active sensing, it is incapable to handle the feature selection task.
>     - Consider label $y$ as the observation of HMM states. In this case, there is no need to perform active sensing and feature selection.
> - Active learning [3]: This approach focuses on label prediction in static settings and is irrelevant to the setup of our active sensing task.
> - Modality selection [4]: This is the most relevant paper. However, it only addresses the feature selection task and assumes predetermined time interval between observations.
> - Fastest change detection [5]: This paper is closely related to optimal stopping which has already been discussed in our manuscript. Further, the correlation structure in sensor networks is unrelated to our active sensing problem.
>
> ## 3. Experiments
> - As requested by the reviewer, we added a new baseline (LL) utilizing the log-likelihood-based reward signal introduced in [4]. The updated benchmark table on the synthetic dataset is provided in the attachment of our global response.
> - Regarding the performance on real-world datasets, we would like to argue that, one important advantage of our method is the optimization for tail-risk samples. As illustrated in Fig. A.1 of the Appendix, our method (RAS) can effectively improve the sensing performance for the tail-risk patients for whom the expected cost $Q^\pi(\mathbf{X})$ is significantly higher under the FO and Fixed baselines.
> - We agree with the reviewer that the inclusion of new real-world datasets for evaluation would help to better justify the performance of our method. We will include new evaluations with the MIMIC-III dataset in the final version of our manuscript.
>
> ## 4. Justification of the synthetic dataset
> In typical application scenarios of active sensing, there exist both variables that are highly predictive about the outcome of interest and those that are noisy, less informative proxy variables about the outcome. The acquisition costs of strong predictors are generally higher than the proxies such that balancing between accuracy and observation costs through active sensing is critical. For the synthetic dataset, variable $x_1$ is the strong predictor of outcome $y$, and $x_2$ is the proxy variable with a higher noise level. Thus, the measurement costs for and are set to be 1.0 and 0.1, respectively. The remaining two variables $x_3$ and $x_4$ are irrelevant to outcome $y$. For reference, their acquisition costs are simply set to 1.0.
>
> ## Reference
>
> [1] Krishnamurthy, V., 2002. Algorithms for optimal scheduling and management of hidden Markov model sensors. IEEE Transactions on Signal Processing, 50(6), pp.1382-1397.
>
> [2] Krishnamurthy, V., 2013. How to schedule measurements of a noisy Markov chain in decision making?. IEEE Transactions on Information Theory, 59(7), pp.4440-4461.
>
> [3] Beyer, C., Büttner, M., Unnikrishnan, V., Schleicher, M., Ntoutsi, E. and Spiliopoulou, M., 2020. Active feature acquisition on data streams under feature drift. Annals of Telecommunications, 75, pp.597-611.
>
> [4] Kossen, J., Cangea, C., Vértes, E., Jaegle, A., Patraucean, V., Ktena, I., Tomasev, N. and Belgrave, D., 2022. Active Acquisition for Multimodal Temporal Data: A Challenging Decision-Making Task. arXiv preprint arXiv:2211.05039.
>
> [5] Heydari, J. and Tajer, A., 2017, March. Quickest change detection in structured data with incomplete information. In 2017 IEEE International Conference on Acoustics, Speech and Signal Processing (ICASSP) (pp. 6434-6438). IEEE.

---

> > ### Comment · Reviewer_d1SC · 2023-08-22
> >
> > I have read the authors' response as well as the rest of the reviews and I have decided to raise my score accordingly. Still I believe that the evaluation is limited.

---

### Official Review · Reviewer_fZr8 · 2023-07-01

**Soundness:** 3 good
**Presentation:** 2 fair
**Contribution:** 3 good
**Rating:** 6
**Confidence:** 2

**Summary:**

This paper proposed an active sensing method that answers the questions of when and what diagnosis test to conduct to optimize the trade-off between the cost of acquisition and the timeliness and accuracy of the predictive model. Compared with the existing active sensing model that assumes a fixed data collection schedule, it builds flexibility to determine the timing of data collection. The paper demonstrates the utility of the proposed using synthetic data and a real-world dataset.

**Strengths:**

The proposed method adds extra flexibility in the decision-making policy to decide when (using continuous time active sensing formulation) to conduct a diagnostic test and what test to complete, which is what the existing model can do. This is a well-motivated problem.


**Weaknesses:**

As shown in the attachment, Table A3 on the empirical experiment with a real-world dataset, the proposed method shows only marginal improvement in accuracy and timeliness, but the cost has almost doubled, as compared with the benchmark; not sure whether these results suggest the better quality of the model overall. This raises questions whether there is practical utility of the proposed method.

In addition,  there are several places the presentation/notation is confusing and hard to follow; see below.
There is some notation confusion - for example, in Figure 2, the arrow from the topic box only sources from the post-interperation sensing history, not the post-interpolation of observational patient trajectory, it is unclear why the exclusion of trajectory history is justifiable, or it is an error of the drawing?

Figure 1, subplot (a) is extremely small; subplot (b), the vertical dot line is not explained.

The equation number and figure number at someplace are not clearly identified, leaving the reader to guess, which cause confusing interpretation, e.g. Page 5 bottom part





**Questions:**

In real-world data set evaluation, it seems to be a three-class classification, while the performance is reported using AUC, etc. which is binary classifier performance metrics, do you weight the multi-class classification in presenting the result? Why it needs to be formulated as classification, is it more relevant if formulating as regression problem instead bin them into discrete categories?



**Limitations:**

author could comment on how the bias/quality of the decision in the original dataset may impact the validity of the approach.

---

> ### Author Rebuttal · Authors · 2023-08-10
>
> We appreciate the reviewer for helping review our paper and providing valuable comments to improve our manuscript. We address the reviewer's concerns on the clarity of our paper and the experimental evaluation of our method as follows.
>
>
> ## 1. Clarity
> We apologize for the confusions. We have fixed the readability and equation numbering issues mentioned by the reviewer and will further improve the clarity of our manuscript in the final version.
>
> ### 1.1 Data flow from observational patient trajectory in Fig. 2
> The reviewer is correct. Data flows should come from both the sensing history and observational patient trajectory. One arrow was missed after the box of interpolator in Fig. 2 of our manuscript. We have corrected this mistake and improved the general readability of Fig 2 in the revision.
>
>
> ## 2. Experiments
>
> ### 2.1 Benchmark in real-world dataset
> In our benchmark, both prediction accuracy (ROC, PRC) and acquisition cost (COST) are important performance metrics to take into account. With focuses on different perspectives of a sensing strategy, there could be multiple ways to rank the four methods in Table 2 of our manuscript. As a balanced score function, we propose criterion $\omega$ as the decrease in accuracy per unit reduction in acquisition cost, i.e., $\omega = \frac{\Delta PRC}{\Delta COST}$, where
>
> - $\Delta PRC = max(0, PRC_{FO} - PRC)$: drop in accuracy, $PRC_{FO}$ is the PRC score of the FO baseline.
> - $\Delta COST = COST_{FO} - COST$: reduction in acquisition cost, $COST_{FO}$ is the COST of the FO baseline.
>
> In Table A3 of the Appendix, our proposed method RAS achieves nearly no decrease in accuracy while reducing the acquisition cost from 26.4 to 9.89. Based on the criterion $\omega$, RAS has the best performance. In comparison, ASAC would be the best method when our main target is to simply reduce the acquisition cost. In practice, the criterion for ranking different approaches should be carefully designed based on the specific application scenario.
>
> Regarding the practical utility of our proposed method, we would like to highlight that, one important advantage of our method is the optimization for tail-risk samples. As illustrated in Fig. A.1 of the Appendix, RAS can effectively improve the sensing performance for the tail-risk patients for whom the expected cost $Q^\pi(\mathbf{X})$ is significantly higher under the FO and Fixed baselines.
>
>
> ### 2.2 AUC score for multi-class classification
> The reviewer is correct. There are three classes in our considered ADNI dataset. For the accuracy evaluation, we apply the one-vs-the-rest (OvR) strategy to calculate the AUC score for each class and report the average scores across all classes as the summary.
>
> ### 2.3 Classification v.s. regression on ADNI
> On the ADNI dataset, we create patient labels from their cognitive test scores (CDRSB) as the indicator of different stages of the disease. As the reviewer mentioned, the cognitive test scores can be directly used in a regression task. However, we would like to argue that the discrete labels are more appropriate descriptions of the disease progression. According to Table 5 in [1], patients with Dementia conditions may have CDRSB scores in a wide range (4.5 to 18.0). This suggests that the CDRSB scores could be highly noisy and are unsuitable for patient staging on the ADNI dataset.
>
>
> ## 3. Discussion of limitations
> We appreciate the reviewer for the comments on decision biases. Decisions in the data collection process of a dataset indeed have a potential impact on our proposed approach. First, the validity of our method relies on the baseline predictor $f_P$. If the training dataset is of low quality, $f_P$ is likely to have poor predictive power, which can compromise the usefulness of our proposed sensing policy. In the meantime, highly biased decisions during data collection may lead to high missing rates of certain feature variables, which could mislead our method to overlook the importance of such features. We will include a detailed discussion in the final version of our manuscript.
>
> ## Reference
>
> [1] O'Bryant SE, Lacritz LH, Hall J, et al. Validation of the new interpretive guidelines for the clinical dementia rating scale sum of boxes score in the national Alzheimer's coordinating center database. Arch Neurol. 2010;67(6):746-749.

---

> > ### Comment · Reviewer_fZr8 · 2023-08-12
> >
> > Thank you for the clarification on the practical utility of the proposed approach. If the optimization for tail-risk samples is important, then it is suggested to include in the main part of the paper, rather than buried in appendix. Those discussion needs to be upfront.

---

> > > ### Author Response · Authors · 2023-08-13
> > > **Thank you for your valuable comments**
> > >
> > > Dear reviewer,
> > >
> > > We sincerely appreciate your valuable comments. We agree that the optimization for tail-risk samples in our proposed approach should be highlighted in our main manuscript. In the final version of our paper, we will include results and discussion on this aspect utilizing the additional one page.
> > >
> > > Given that we have addressed your primary concerns raised in the review, we kindly ask you to consider adjusting the review score while taking our rebuttal into account.
> > >
> > > Your comments have greatly helped us improve the clarity of our manuscript, and we are genuinely grateful for your thoughtful suggestions. Thank you once again for your time and input.
> > >
> > >
> > > Best regards,
> > >
> > > Paper Authors

---

### Official Review · Reviewer_mJK8 · 2023-07-07

**Soundness:** 3 good
**Presentation:** 3 good
**Contribution:** 3 good
**Rating:** 6
**Confidence:** 3

**Summary:**

This work studies the problem of active cost-aware feature acquisition assuming time varying feature settings via breaking-down feature selection and prediction decision making as two policies. The problem being considered is generally a difficult problem even with the non time-varying feature settings.

I have a few questions about the real-world applicability, assumptions, and the approach. I would like to hear from the authors (see below).

**Strengths:**

The problem is well motivated for healthcare domain and is of clear practical value.

The paper is well-written and clear

I found considering risk and long-tail behavior very interesting and quite important as cost optimization often means dropping precision for minorities which is not acceptable for health domain usecases


**Weaknesses:**

The method used to optimize the cost of acquisition i.e., a linear coefficient to balance cost vs. prediction risk is quite simple and does not necessarily lead to stable or near-optimal results. In the real-world settings the balance point might be different for each sample, and it is even more complicated when we take the time-varying assumption into account.

(see other comments)

**Questions:**

I was wondering what was the motivation behind having two policies rather than one with larger action space?

Following on the previous question, wouldn’t the two-policy approach result in less optimal solutions compared to the optimization of one joint policy?

In the real-world settings that we do not have ground-truth values train set (assuming a cost-aware approach or a version of your method was applied at the time of data collection) how would the training process work? Please correct me but my understanding is that we do need fully observed train set for building the baseline predictor.

**Limitations:**

I have no particular concern

---

> ### Author Rebuttal · Authors · 2023-08-10
>
> We sincerely appreciate the reviewer's valuable comments and suggestions. We address the questions on our problem formulation and experiments as follows.
>
> ## 1. Problem formulation
>
> ### 1.1 Motivation of policy decomposition
> The motivation for our sensing policy decomposition is three folds:
>
> - First, the action spaces of the acquisition scheduler and the feature selector are orthogonal and can generate all possible decisions of "what to observe, and when" at the next visit given patient trajectory.
> - Second, for each feature variable, the following decision cycle happens repeatedly: i) new acquisition of patient covariates is performed based on the latest sensing history; ii) then, the next follow-up time is determined based on the new sensing history including the newly accrued measurements in step i). The policy decomposition of acquisition scheduler $\pi_\Delta$ and feature selector $\pi_m$ is a straightforward reflection of such a practical sensing cycle.
> - Finally, in most real-world scenarios, multiple selected feature variables are usually measured at the same visit. Thereby, we combine the decisions on individual feature selection at each visit into a binary vector $\mathbf{m}$ and model its distribution with feature selector $\pi_m$.
>
>
>
> ### 1.2 Optimality of solution
> As we explained above, the two sub-policies in our formulation have orthogonal action space and are mathematically equivalent to a global policy that determines observation interval $\Delta$ and feature selection mask $\mathbf{m}$ simultaneously. Despite the policy decomposition, the solution of our proposed active sensing problem shall be equivalent to the optimal global policy with a larger action space.
>
>
> ## 2. Method
>
> ### 2.1 Balance between the acquisition cost and prediction accuracy
> We agree with the reviewer that, in some real-world settings, the balance point between the feature acquisition cost and prediction accuracy may vary across different samples, and the linear equilibrium achieved under our objective function may not be sufficient to tackle such a scenario. However, our method can be easily extended to tackle such challenges. For instance, by varying the linear coefficient $\lambda$ in some reasonable range, a list of optimal sensing policies can be obtained with our proposed method. Then, we can construct an ensemble of these policies and fine-tune the weights for different policies on an individual sample basis. This approach should be applicable even when the balance between acquisition cost and prediction accuracy is time-dependent. We leave this as our future work as it is out of the scope of our manuscript.
>
> ### 2.2 Training set with missing values
> The reviewer is correct that a baseline predictor is required to train our proposed sensing approach RAS. Leveraging the power of neural CDE in tackling missing data, the baseline predictor can be built on training data with partial observations. Thus, the training set does not need to be fully observed (e.g., MIMIC-III). Indeed, we note that the effectiveness of RAS is highly correlated with the quality of baseline predictor $f_P$. A training set with high missing rates in feature records will generally lead to poor accuracy of $f_P$, which will unavoidably affect the validity of our sensing policy. However, it is worth highlighting that the same baseline predictor was used for the benchmarks (i.e., the variations of our model).
>
> ### 2.3 Online training during data collection
> Our method is designed to work with batch data in offline settings. It should not be trained online during data collection time.

---

> > ### Comment · Reviewer_mJK8 · 2023-08-12
> > **Re: Rebuttal by Authors**
> >
> > I have read the authors feedback and updated my evaluation accordingly. I appreciate the response to my questions but I still have concerns the practical application of this method.

---

> > > ### Author Response · Authors · 2023-08-16
> > > **We appreciate your insightful comments**
> > >
> > > Dear reviewer,
> > >
> > > We appreciate your insightful comments. We would like to further clarify on the practical application of our method.
> > >
> > > First, we concur with the reviewer's observation that our approach, which maintains a linear trade-off between diagnostic accuracy and observation costs, has some limitations. In complex settings where a personalized trade-off coefficient is imperative, our method may not generate the optimal solution. Nevertheless, we contend that the utilization of a similar linear scalarization technique has found extensive application in solving practical multi-objective optimization problems [1]. Within the specific scenarios outlined in our manuscript, our method adeptly attains both stability and optimality through the linear trade-off approach. We acknowledge that extending our method to accommodate tasks necessitating an individualized trade-off is an important area for future exploration.
> > >
> > > Second, regarding the practical application of our method, we have consulted with several clinical collaborators. They consider our objective function (as a weighted sum of accuracy and costs) to be a reasonable design choice for real-world applications. Furthermore, our clinical collaborators have identified three distinct practical scenarios wherein our method is exceptionally well-suited:
> > >
> > > 1. Active surveillance for prostate cancer [2]. Prostate cancer is a common disease for male patients and overdiagnosis is a problem. Most individuals newly diagnosed with low-grade prostate cancer would not require immediate treatment (radiotherapy, surgery etc.) which has a great impact on their life quality. Active surveillance is the only approach to pinpoint patients with higher risks of death due to prostate cancer.
> > > 2. Lung nodule management in lung cancer screening. The process of lung cancer screening frequently detects lung nodules, the malignancy of which remains uncertain. Empirical risk models alongside repeated CT/PET scans are widely adopted to monitor the disease progression. However, definitive diagnosis often necessitates invasive procedures like biopsy or surgical resection, which pose risks to patients. In this context, the consideration of additional diagnostic accuracy of repeated scanning weighed against cost (including radiation exposure) in active sensing is valuable.
> > > 3. The ongoing management of cancer recurrence. Cancer recurrence may occur weeks or even years following the initial treatment. Early detection and intervention of the recurrence mandates vigilant monitoring. However, the predictive variables involved—biomarkers, imaging, biopsies—carry concomitant costs, radiation exposure, and potential adverse consequences for patients. Notably, radiation exposure is established as a significant risk factor for second, independent cancers among those treated for cancer [3]. Here, active sensing is essential in minimizing the adverse impact on patients during the management of cancer recurrence.
> > >
> > > These clinical scenarios underscore the real-world relevance and practical impact of our work. We trust that these examples more effectively highlight the significance of our approach.
> > >
> > > We sincerely hope that this response adequately addresses your concerns regarding the practical applications of our method. Once again, we extend our gratitude for your time and valuable comments.
> > >
> > > Best regards,
> > >
> > > Paper Authors
> > >
> > >
> > > [1] Eriskin, Levent, Mumtaz Karatas, and Yu-Jun Zheng. "A robust multi-objective model for healthcare resource management and location planning during pandemics." Annals of Operations Research (2022): 1-48.
> > >
> > > [2] https://www.nice.org.uk/guidance/ng131/chapter/Recommendations.
> > >
> > > [3] Demoor-Goldschmidt, Charlotte, and Florent de Vathaire. "Review of risk factors of secondary cancers among cancer survivors." The British journal of radiology 92.1093 (2019): 20180390.

---

### Official Review · Reviewer_L7Er · 2023-07-27

**Soundness:** 3 good
**Presentation:** 2 fair
**Contribution:** 3 good
**Rating:** 3
**Confidence:** 3

**Summary:**

This paper investigates timely outcome prediction by proposing a novel risk-averse active sensing approach RAS. The proposed RAS decomposes the policy into acquisition scheduler and feature selector to address the composite decision problem of when to conduct the acquisition and which measurements to make. In addition, RAS enables the prioritization of tail-risk patients in the risk-aversion training procedure. The experiments on synthetic and real-world healthcare datasets show its effectiveness.

**Strengths:**

This paper is studying a very interesting and important problem - active sensing methods for early detection and intervention of adverse events. The proposed risk-averse active sensing method is technically sound. The experimental results show its effectiveness compared to the three baselines.

**Weaknesses:**

Minor comments: 1.there are so many symbols used in the paper writing, it would be helpful to include a notation table in the appendix. 2. The titles of Section 2 and Section 3 are confusing. Probably, Section 2 is Problem Formulation (or General Framework), Section 3 is Methodology. 3. Highlight the best performance in Table 2.

**Questions:**

The result shown in Figure 3 shows that both NRAS and RAS can reduce the long tail in the distribution. It seems NRAS is better than RAS for reducing the long tail, right? Why does this happen?

**Limitations:**

The authors have discussed broader impact of this work.

---

> ### Author Rebuttal · Authors · 2023-08-10
>
> We thank the reviewer for the insightful comments. The reviewer’s comments regarding clarity and experiment results are addressed below.
>
> ### 1. Notation table.
>
> We thank the reviewer for the suggestion of including a notation table to improve clarity. The definition and explanation of major notations used in our manuscript are listed in Table R1.
>
> Table R1. Major notations in the manuscript.
>
> | Notation     | Definition                               | Notation               | Definition                                |
> | ----------   | ------------                             | --------------------   | ----------------------                    |
> | $\mathbf{x}$ | Patient feature variable                 | $\mathbf{X}$           | Patient trajectory                        |
> | $y$          | Patient outcome                          | $\mathbf{\hat{X}}^\pi$ | Sensing history under policy $\pi$        |
> | $\pi$        | Sensing policy                           | $\pi_\Delta$           | Acquisition scheduler                     |
> | $\pi_m$      | Feature selector                         | $\mathbf{m}$           | Selection mask                            |
> | $D_{JS}$     | Jensen–Shannon divergence                | $\mathbf{c}$           | Cost vector                               |
> | $r_y$        | Cumulative mismatch in label prediction  | $r_m$                  | Cost of feature acquisition               |
> | $r$          | Reward signal                            | $Q^\pi(\mathbf{X})$    | Expected cost                             |
> | $CVaR$       | Conditional value-at-risk                | $\alpha$               | Tail-risk quantile                        |
> | $T$          | End time of observation                  | $I$                    | Number of observations in sensing history |
> | $N$          | Number of samples                        | $\mathcal{I}$          | Interpolator                              |
>
>
> We will include a more detailed notation table in the final version of our paper.
>
> ### 2. Titles of Section 2 and 3
>
> We apologize for the confusion caused by the similar titles used for Section 2 and 3 in our manuscript. We will modify the title of Section 2 to "Problem Formulation," as recommended by the reviewer.
>
> ### 3. Highlight the best performance in Table 2.
> In our benchmark, both prediction accuracy (AROC, PRC) and acquisition cost (COST) are important performance metrics considering their clinical impacts. Focusing on various perspectives of a sensing approach, there could be multiple ways to rank the four methods in Table 2. Here, we propose one possible criterion, denoted as $\omega$, which measures the decrease in accuracy per unit reduction in acquisition cost, i.e., $\omega = \frac{\Delta PRC}{\Delta COST}$, where
>
> - $\Delta PRC = max(0, PRC_{FO} - PRC)$: drop in accuracy, $PRC_{FO}$ is the PRC score of the FO baseline.
> - $\Delta COST = COST_{FO} - COST$: reduction in acquisition cost, $COST_{FO}$ is the COST of the FO baseline.
>
> Note that our method, RAS, achieves no loss in accuracy while reducing the acquisition cost from 39.6 to 4.535. Based on the criterion $\omega$, RAS provides the best performance which can be highlighted in Table R2 below:
>
> Table R2: Benchmark of active sensing performance.
>
> | METHOD  | ROC         | PRC         | COST         | $d_{δ=0.3}$ | $d_{δ=0.5}$ | $d_{δ=0.7}$ |
> | ------  | ----        | ---         | ---          | ----        | ---         | ---         |
> | FO      | 0.668±0.000 | 0.634±0.000 | 39.600±0.000 | 0.582±0.000 | 0.229±0.000 | 0.181±0.000 |
> | ASAC    | 0.582±0.035 | 0.527±0.023 | 9.189±1.895  | 1.052±0.339 | 1.326±0.063 | 1.323±0.065 |
> | FIXED   | 0.655±0.006 | 0.600±0.005 | 0.907±0.034  | 1.384±0.000 | 1.398±0.000 | 1.359±0.000 |
> | **RAS** | 0.678±0.006 | 0.635±0.005 | 4.535±0.088  | 0.142±0.018 | 0.132±0.028 | 0.154±0.032 |
>
> ### 4. Comparison of total acquisition cost distribution in Figure 3.
> We appreciate the reviewer for pointing this out. As mentioned by the reviewer, both RAS and its non-risk-averse version (NRAS) can mitigate the long tail in the distribution of the expected cost function $Q^\pi(\mathbf{X})$ compared to the baseline policy with a constant decision interval. Due to the inherent randomness of the stochastic policies, in a single evaluation, the NRAS baseline might have appeared to outperform RAS at some outliers as observed in Figure 3. For a more clear comparison between RAS and NRAS, we have reconducted the experiment in Figure 3 with the average cost $Q^\pi(\mathbf{X})$ of each sample averaged over 5 random training/testing splits. The new result clearly demonstrates that our method RAS can further reduce the long tail in $Q^\pi(\mathbf{X})$ distribution compared to NRAS, highlighting the importance of our proposed tail-risk minimization strategy. Please check the PDF attachment for the updated figure.

---

> > ### Comment · Reviewer_L7Er · 2023-08-16
> > **Thanks for your response**
> >
> > Thanks for the response. I have read it. The new experimental result is quite different with the result shown in the submitted paper. This makes me concern about the reliability of the proposed method and experiments in this paper.

---

> > > ### Author Response · Authors · 2023-08-17
> > > **We apologize for any lack of clarity in describing the new results in our rebuttal**
> > >
> > > Dear Reviewer,
> > >
> > > We apologize for the confusion regarding the new results we posted in our rebuttal. We would like to provide further clarification on the changes.
> > >
> > > ### 1. Updated Table 1 in the attachment
> > >
> > > As requested by Reviewer 1dSC, we have included a new baseline (LL) in the benchmark. All other results in Table 1 remain unchanged. Our analysis remains effective with the addition of the new baseline.
> > >
> > > ### 2. New results in Figure 3
> > >
> > > We believe there may have been a misunderstanding by the reviewer due to our unclear description in the rebuttal. The new results are still consistent with our original Figure 3. The differences in the results are due to slight changes in the setup, which were necessary given the limited time available for the rebuttal period.
> > >
> > > - Since our primary focus is to highlight the effectiveness of RAS in reducing long-tail distribution in $Q^\pi(\mathbf{X})$, we chose to conduct the experiment for Figure 3 with a smaller number of training epochs to facilitate timely evaluation. As illustrated in the updated Figure 3 in the attached document, RAS effectively reduces the number of tail samples compared to NRAS, even with fewer training epochs in the new setup.
> > > - It is important to note that the convergence speed of NRAS and RAS differs due to the reduced number of effective samples (only tail-risk samples) for RAS in each epoch. The reduction in the number of training epochs unavoidably affected the shape of $Q^\pi(\mathbf{X})$ distributions in the new results. We apologize for any confusion and assure you that a sufficient number of training epochs will be used in the final version of our manuscript.
> > > - In the new results, we plotted all samples in the synthetic dataset to emphasize the difference between RAS and NRAS in the long tail. In contrast, only test set samples were used in the original Figure 3. This led to changes in the y-axis scales and potentially impacted the shape of the $Q^\pi(\mathbf{X})$ distribution as well.
> > > - The Fixed baseline was not included in the new Figure 3, which resulted in a different scale on the x-axis.
> > >
> > > Once again, we extend our gratitude to the reviewer for the thorough evaluation of our new results. We apologize for any lack of clarity in describing the new setup in our rebuttal. We hope that our clarifications can address the reviewer's concerns regarding the new results and reliability of our method and lead to a reconsideration of the reviewer's score in our rebuttal.
> > >
> > > Best regards,
> > >
> > > Paper Authors

---

> > > > ### Author Response · Authors · 2023-08-22
> > > > **Follow-up on the reviewer's concerns**
> > > >
> > > > Dear Reviewer,
> > > >
> > > > We sincerely appreciate your meticulous review of our manuscript and the concerns you raised regarding the new results. We have provided detailed clarification on the new experimental setup and explained the discrepancy in the results. If you find that our response has resolved your concerns, we humbly request a reconsideration of the review score.
> > > >
> > > > Thank you for your thoughtful consideration.
> > > >
> > > > Best regards,
> > > >
> > > > Paper Authors

---

### Author Rebuttal · Authors · 2023-08-10

We would like to thank all reviewers for taking the necessary time and effort to review our manuscript. We sincerely appreciate all your valuable comments and suggestions, which helped us in improving the quality of the manuscript.


## Summary of related work mentioned by reviewer d1SC
We thank reviewer d1SC for providing an extensive list of pertinent literature. We put a summarized table of these papers in Table R1 below for reference.

Table R1. Comparison with additional literature.

| Method     | Focus                 | Observation                           | Time interval         | Action                                                       |
| ------     | -------               | ---------                             | -----------           | --------------                                               |
| RAS (Ours) | Label $y$     | Sensing history $\tilde{\mathbf{X}}$  | Adaptive (continuous) | Determine when and what to observe                           |
| [1,2,4]    | HMM state $e$         | Discrete measurement of $e$           | Constant              | Select most relevant sensor to measure $e$                   |
| [3]        | HMM state $e$         | Scalar measurement of $e$             | Adaptive (discrete)   | Schedule measurement to $e$                                  |
| [5]        | HMM state $e$         | Observation of $e$                    | Constant              | Track the hidden state $e$                                   |
| [6]        | HMM state $e$         | Location of $e$                       | Constant              | Schedule sensors to locate a target object                   |
| [7]        | Model accuracy        | Sample stream                         | Constant              | Select sample and features to improve training set    |
| [8]        | Label $y$             | multi-modal data $\mathbf{\tilde{x}}$ | Constant              | Query most relevant modality data source for prediction      |
| [9,10]     | Correlation structure | Node states                           | Constant              | Detect the change of correlation structure in sensor network |



## Reference

[1] Krishnamurthy, V., 2002. Algorithms for optimal scheduling and management of hidden Markov model sensors. IEEE Transactions on Signal Processing, 50(6), pp.1382-1397.

[2] Krishnamurthy, V. and Djonin, D.V., 2007. Structured threshold policies for dynamic sensor scheduling—A partially observed Markov decision process approach. IEEE Transactions on Signal Processing, 55(10), pp.4938-4957.

[3] Krishnamurthy, V., 2013. How to schedule measurements of a noisy Markov chain in decision making?. IEEE Transactions on Information Theory, 59(7), pp.4440-4461.

[4] Zois, D.S. and Mitra, U., 2017. Active State Tracking With Sensing Costs: Analysis of Two-States and Methods for n-States. IEEE Transactions on Signal Processing, 65(11), pp.2828-2843.

[5] Molloy, T.L. and Nair, G.N., 2021, June. Active trajectory estimation for partially observed Markov decision processes via conditional entropy. In 2021 European Control Conference (ECC) (pp. 385-391). IEEE.

[6] Atia, G.K., Veeravalli, V.V. and Fuemmeler, J.A., 2011. Sensor scheduling for energy-efficient target tracking in sensor networks. IEEE Transactions on Signal Processing, 59(10), pp.4923-4937.

[7] Beyer, C., Büttner, M., Unnikrishnan, V., Schleicher, M., Ntoutsi, E. and Spiliopoulou, M., 2020. Active feature acquisition on data streams under feature drift. Annals of Telecommunications, 75, pp.597-611.

[8] Kossen, J., Cangea, C., Vértes, E., Jaegle, A., Patraucean, V., Ktena, I., Tomasev, N. and Belgrave, D., 2022. Active Acquisition for Multimodal Temporal Data: A Challenging Decision-Making Task. arXiv preprint arXiv:2211.05039.

[9] Heydari, J. and Tajer, A., 2017, March. Quickest change detection in structured data with incomplete information. In 2017 IEEE International Conference on Acoustics, Speech and Signal Processing (ICASSP) (pp. 6434-6438). IEEE.

[10] Chaudhuri, A., Fellouris, G. and Tajer, A., 2021, July. Sequential change detection of a correlation structure under a sampling constraint. In 2021 IEEE International Symposium on Information Theory (ISIT) (pp. 605-610). IEEE.

---

### Decision · Program_Chairs · 2023-09-21

**Decision:**

Accept (poster)

**Comment:**

This paper has been assessed by four knowledgeable reviewers whose recommendations included: weak acceptance (2), borderline acceptance (1) and straight rejection (1), respectively. The main complaints from the reviewers include limited evaluation, concerns about the practical utility of the presented method, important related work - perhaps not directly but conceptually relevant - missing in the discussion and comparisons, and poor organization of the paper whereas some key concepts were relegated to the appendix preventing the main paper to stand firmly on its own. The authors provided rebuttals that alleviated some but not all of the qualms. However, all things considered, and setting aside the most negative but also very curt review short of meaningful arguments, this paper is worth a serious consideration for acceptance for NeurIPS.